# Role of Artificial Intelligence in Radiogenomics for Cancers in the Era of Precision Medicine

**DOI:** 10.3390/cancers14122860

**Published:** 2022-06-09

**Authors:** Sanjay Saxena, Biswajit Jena, Neha Gupta, Suchismita Das, Deepaneeta Sarmah, Pallab Bhattacharya, Tanmay Nath, Sudip Paul, Mostafa M. Fouda, Manudeep Kalra, Luca Saba, Gyan Pareek, Jasjit S. Suri

**Affiliations:** 1Department of Computer Science & Engineering, International Institute of Information Technology, Bhubaneswar 751003, India; sanjay@iiit-bh.ac.in (S.S.); c118002@iiit-bh.ac.in (B.J.); suchismita.dasfcs@kiit.ac.in (S.D.); 2Department of Information Technology, Bharati Vidyapeeth College of Engineering, New Delhi 110063, India; neha.gupta@bharatividyapeeth.edu; 3National Institute of Pharmaceutical Education & Research, Ahmedabad 382355, India; deepaneeta.sarmah@niperahm.res.in (D.S.); pallab.bhattacharya@niperahm.res.in (P.B.); 4Department of Biostatistics, Johns Hopkins University, Baltimore, MD 21218, USA; tnath3@jhu.edu; 5Department of Biomedical Engineering, North Eastern Hill University, Shilong 793022, India; spaul@nehu.ac.in; 6Department of Electrical and Computer Engineering, Idaho State University, Pocatello, ID 83209, USA; mfouda@isu.edu; 7Department of Radiology, Massachusetts General Hospital, Boston, MA 02114, USA; mkalra@mgh.harvard.edu; 8Department of Radiology, A.O.U., di Cagliari-Polo di Monserrato s.s., 09124 Cagliari, Italy; lucasaba@tiscali.com; 9Minimally Invasive Urology Institute, Brown University, Providence, RI 02912, USA; gyan_pareek@brown.edu; 10Stroke Monitoring and Diagnostic Division, AtheroPoint™, Roseville, CA 95661, USA; 11Knowledge Engineering Center, Global Biomedical Technologies, Inc., Roseville, CA 95661, USA

**Keywords:** radiogenomics, cancer, oncology, artificial intelligence, machine learning, deep learning

## Abstract

**Simple Summary:**

Recently, radiogenomics has played a significant role and offered a new understanding of cancer’s biology and behavior in response to standard therapy. It also provides a more precise prognosis, investigation, and analysis of the patient’s cancer. Over the years, Artificial Intelligence (AI) has provided a significant strength in radiogenomics. In this paper, we offer computational and oncological prospects of the role of AI in radiogenomics, as well as its offers, achievements, opportunities, and limitations in the current clinical practices.

**Abstract:**

Radiogenomics, a combination of “Radiomics” and “Genomics,” using Artificial Intelligence (AI) has recently emerged as the state-of-the-art science in precision medicine, especially in oncology care. Radiogenomics syndicates large-scale quantifiable data extracted from radiological medical images enveloped with personalized genomic phenotypes. It fabricates a prediction model through various AI methods to stratify the risk of patients, monitor therapeutic approaches, and assess clinical outcomes. It has recently shown tremendous achievements in prognosis, treatment planning, survival prediction, heterogeneity analysis, reoccurrence, and progression-free survival for human cancer study. Although AI has shown immense performance in oncology care in various clinical aspects, it has several challenges and limitations. The proposed review provides an overview of radiogenomics with the viewpoints on the role of AI in terms of its promises for computational as well as oncological aspects and offers achievements and opportunities in the era of precision medicine. The review also presents various recommendations to diminish these obstacles.

## 1. Introduction

Cancer is a second leading cause of death worldwide, right after cardiovascular diseases, accounting for nearly 10 million deaths in 2020. As per world health organization (WHO) statistics, the common types of cancers that people suffer more are those of the breast, lungs, colorectal, prostate, skin, brain, and stomach [1]. The cancer burden continues to grow globally, exerting tremendous physical, emotional, and financial strain on individuals, families, communities, and health systems. Countries with mediocre and poor health infrastructure do not have the access to timely, quality diagnosis and treatment for a large number of patients [2].

In the era of precision medicine, molecular characterization of cancer using genomic technology is essential [3,4]. In the last few years, significant progress has been observed in molecular characterization. However, due to the technical complexity, cost, and turnaround time, a vast scale genome-based characterization of cancer is not yet routinely adapted for all types of cancers [5,6,7]. In existing clinical practices, due to the heterogeneous behavior of cancer, molecular profiling is often limited, and heterogeneity of the tumor is repeatedly missed when a portion of the cancer is examined [8]. Throughout the treatment, determination of molecular targets requires ex vivo postoperative analysis of the resected tumor or biopsy sample. This has restricted the assessment of tumors’ spatial and temporal heterogeneity and is not possible to determine the molecular transformation of cancer continuously [9]. Additionally, in the case of the solid type of tumors, the functional, anatomic, and physiological properties of the whole tumor may not be fully reflected in the histopathological samples [10,11]. Researchers and scientists worldwide have noticed the substantial job of medical imaging in clinical treatment decision-making and in analyzing cancers [12]. Earlier, its main job was restricted to prognosis and staging [13]. However, recently, imaging-derived markers obtained from clinical images have significantly been investigated to deliver insight into cancer non-invasively. Most importantly, imaging helps characterize the peritumoral regions, as these regions are not always invasively removed for the molecular characterization of cancer [14,15].

Recently, radiogenomics, the combination of “*Radiomics*” and “*Genomics*,” has significantly drawn the attention of researchers to determine *imaging surrogates* for genomic signatures and to advance biomarkers leveraging the numerous data types used to characterize cancer. These biomarkers can be used in different clinical decision-making such as survival prediction, tumor progression, reoccurrence, and heterogeneity analysis.

In “*Radiomics*,” different quantitative medical imaging features are extracted computationally that capture different imaging phenotypes; these are not easily noticed with the uncovered eye [16]. Recent research demonstrated that cancer’s molecular information is linked with the imaging phenotype [17]. The basic, fundamental step in radiomics includes image data acquisition, preprocessing of image data such as filtering [18], region of interest (ROI), segmentation [19,20], different types of features extraction [21,22], and then use of these extracted features for appropriate analysis. ROI extraction in the imaging data must be performed manually or semi/fully automatically using computational algorithms approved by the expert neuropathologist (neurooncologist). Quantitative features extraction includes different features such as histogram-based, first, second, or higher-order features [23,24,25,26]. Recently, high-level features obtained from deep learning have also been significantly used to analyze cancer regions [27,28,29]. In “*Genomics*”, the human genome is examined to analyze cancer by extracting several genomic features (genotypes). The genotype basically presents the genetic information of cancer.

Further, these obtained radiomics and genomics features are used by different AI-based methods to characterize and analyze cancer. In recent years, AI has presented data-driven examination models that have managed the noteworthy signs of progress in information-processing methods in radiogenomics of cancer. There have been constant and incremental determinations to improve AI’s analytic efficiency to be endorsed for clinical practice [30,31]. The discovery of artificial neural networks (ANN) and their subsequent development [32,33] presents computational learning models: machine- and deep-learning ideologies. It is mainly accountable for the development of AI in the field of radiogenomics.

In the proposed article, our main objective is to provide and discuss different perspectives regarding the contemporary and inherent responsibility of AI methods in radiogenomics of cancer, including current challenges and prospects. We will start this article by providing an insight into radiogenomics and its achievements. Further, we will discuss opportunities provided by AI and how it is significantly used in different recent studies of cancers by providing an analytical form of investigation. In the end, we will conclude with the overall perspective of using AI in radiogenomics of cancer, applied in clinical decision-making in the epoch of individualized medicine and care.

## 2. Search Strategy and Statistics of Radiogenomics Studies

This section deals with the search strategy using the Preferred Reporting Items for Systematic Reviews and Meta-Analyses (PRISMA) model, followed by statistical distribution and analysis of various radiogenomics studies.

### 2.1. The PRISMA Model

The proposed narrative review has basically been designed to analyze the role and impact of AI in radiogenomics study. The PRISMA model has been adapted for this purpose, as shown in Figure 1. A detailed search was performed using top academic search databases such as Google Scholar, PubMed, IEEE Xplore, ScienceDirect, Springer, and MDPI. The relevant keywords used for the search included “radiogenomics”, “radiomics”, “genomics”, radiogenomics using AI”, “machine learning for radiogenomics”, and “deep learning for radiogenomics”. A total of 154 records were collected. The duplicate records were removed from this collection using the “Find Duplicates” feature in EndNote software by Clarivate Analytics, which resulted in 104 articles. The three exclusion criteria (marked as E1, E2, and E3 in Figure 1) removed 23, 20, and 10 articles under the category of (i) non-relevant articles, (ii) studies not related to AI, and (iii) articles with insufficient data. Finally, 51 relevant articles were used for the qualitative synthesis to know the impact and role of AI in radiogenomics studies.

### 2.2. Statistical Distributions of AI Attributes of Radiogenomics Studies

#### 2.2.1. Statistical Distribution of Publication Trends of Radiogenomics Using AI

Since radiogenomics is a new domain of study in cancer research, the number of publications in the initial stages is low; however, it has continued to grow over the past few years. It has been emerging for the past five years, as shown in Figure 2a. However, the number of publications in the form of search is low, and it is expected to increase in the recent future as it is applicable to all kinds of cancer research, providing an extra edge over the other methodologies.

#### 2.2.2. Statistical Distribution of Country-Wise Study of Radiogenomics Using AI

As this is a current and trending topic in the research of the deadliest cancerous disease, many research publications have been published across the globe. As it is an emerging domain of research in oncology, there is evident curiosity as to the leading contributors in terms of country. Figure 2b depicts the pie-chart distribution of county-wise research publications on the set of radiogenomics studies we have considered. It shows that the USA and China are the leading contributors, with maximum percentages of 39% and 31%, respectively.

#### 2.2.3. Statistical Distribution of AI and Its Model Used in Radiogenomics Studies

Artificial Intelligence has been successfully serving every domain of computer vision applications in the healthcare industry. AI has also helped radiogenomics studies to become automotive. Both machine learning and deep learning under the umbrella of AI take radiogenomics studies to a further level, with precision in performances. Under ML, traditional radiomics features are extracted, while under DL models, the automatic deep features help the AI model better classify the status of genomics in radiology. Our finding indicates that ML has been used a bit more compared to DL, which is shown in Figure 3a. Similarly, the frequency of various ML and DL models that have been used for this proposal, including convolutional neural network (CNN), regression, random forest (RF), support vector machine (SVM), ResNet, XGBoost, VGG, naïve Bayes, artificial neural network, DenseNet, GoogleNet, K-NN, decision tree, and linear discriminant analysis (LDA), is shown in Figure 3b.

#### 2.2.4. Statistical Distribution of Image Modalities Used in Radiogenomics

MRI, CT, and PET are the prominent imaging modalities considered for radiogenomics studies, as shown in Figure 4a. Their distributions are also shown in the pie chart, with their corresponding share in percentages. MRI has a greater share of 45%, which indicates that this modality can be used for all types of anatomical cancer; however, MRI is preferable in brain-tissue characterization. Apart from MRI, CT imaging has also been used largely with a share of 39% for the radiogenomics studies under this review, while others are combinations of MRI, CT, and PET. Additionally, mammography is used as an important image modality of breast cancer.

#### 2.2.5. Statistical Distribution of Anatomical Area of Cancer in Radiogenomics

The various cancer types, according to different anatomical areas considered for this radiogenomics review, have depicted in Figure 4b. Among these cancer types, brain, breast, and lung cancer have been found to be more frequently analyzed for radiogenomics compatibility, with 23%, 14%, and 15%, respectively. However, this field of radiogenomics applies to all types of cancer that developed from the mutation of genes. The other prominent cancer types considered here for radiogenomics include liver, ovarian, collateral, gastric, prostate, kidney, head and neck, and skeletal muscle, as shown in Figure 4b.

#### 2.2.6. Statistical Distribution of Dataset Used in Radiogenomics

The dataset size considered for the radiogenomics studies under this review includes the number of patients considered for the corresponding study. This includes all the objectives and the modality used in the study. As radiogenomics is a relatively new field of research, the dataset is not easily available for public use, even if not volumetric. It is observed that all the studies considered have datasets within nearly 1000 objects, and a few studies have also been limited to below 100 objects, as depicted in Figure 5. A higher data size is expected for better evaluation of a radiogenomics AI system to avoid data imbalance and over-fitting conditions.

#### 2.2.7. Performance Analysis of Radiogenomics Studies

Performance evaluation has been the final and essential part of an AI-based diagnosis system. The higher the values of performance evaluation parameters, the better the AI system. Most of the radiogenomics studies have considered accuracy and area under the receiver-operating characteristic curve (AUC) as suitable performance evaluation parameters. However, sensitivity, specificity, precision, and other methods of statistical evaluation have been partly used. The mean and standard deviation (SD) of the radiogenomics studies were found to have an accuracy, in percentage, (84.34 ± 9.37) and AUC (expressed in percentage, 85.42 ± 7.95), respectively, as shown in Figure 6.

## 3. An Insight of Radiogenomics

The following subsections describe the components of radiogenomics. The entire pipeline is also presented, depicting different modules in radiogenomics.

### 3.1. Conventional and Deep Radiomics

Radiomics deals with the mining and extraction of quantitative medical imaging features that are helpful in clinical assessment methods to expand the prognostic, diagnostic, and predictive precision of disease. Radiomics technology can be applied to medical imaging of any disease; however, it is gaining importance in cancer research for personalized treatments. It has been applied quite successfully for all anatomical areas of cancerous images of multiple modalities such as MRI, CT, PET, US, etc. Each modality has its own peculiarities, while considering the tissue-level radiography of various anatomical sections. The conventional radiomics models primarily depend on explicitly hand-crafted features from radiological images [34,35]. These wide ranges of radiomics features can be texture, geometric, intensity, shape, histogram, dynamics curve, angiogenesis, metabolic, morphological, spatial, and statistical features, and even some high-dimensional features too [36,37,38,39]. Each feature has special importance for defining the imaging phenotype and revealing key components of the tumor phenotype. The most prominent texture features define the pattern and spatial arrangement of colors or intensities of the tumor. The geometric features describe the 3D shape, size, location, and dynamics curve characteristics of the tumorous image. The intensity features demonstrate the pixel or voxel intensities within the tumor image.

However, in recent years, the development of deep-learning technologies in computer vision has attracted the application of radiomics. The automatic feature-extraction of deep radiomics helps find relevant and useful features to extract on a large scale. The deep radiomics features depend upon the network’s depth, with stacks of convolutional and fully connected deep layers [40]. The automatic process of deep radiomics also includes the auto feature-selection process, which may not be available with traditional radiomics. Traditional and deep radiomics find the appropriate phenotype information of cancer to classify the tumor diagnosis, prognosis, and personalized cancer treatments. Figure 7 shows the differences in extracting phenotype information by traditional radiomics and deep-learning methods. The next subsection describes the role and significance of genomics study in cancer research.

### 3.2. Significance of Genomics Study in Cancer Research

Genomic study of cancer is a relatively new area that considers the benefits of the recent advances in technology to examine the human genome, which comprises the entire set of DNA. By comparing DNA and RNA sequencing of cancer cells with the normal tissue, scientists and researchers identify genetic differences which could be the root cause of cancer [41].

Basically, cancer is caused by the unbounded germination of the cancerous cell [42]. DNA is the central control system of the cell with lots of genetic features (genotype) and defines the cell’s behavior. The uncontrolled growth of cells may include DNA mutations, deletions, rearrangements, amplification, and the addition or removal of the chemical mark. The genotype is basically the investigation of the genetic constitution of an individual organism. Some prominent genotype of the human body includes Isocitrate dehydrogenase (IDH), Tumor Protien53 (TP53), epidermal growth-factor receptor (EGFR), O6-methylguanine-DNA methyltransferase (MGMT), etc. [43]. This genomics has a wide-ranging functionality in the human body for complete nourishment and growth. One example is TP53, a genomics protein that helps with DNA repair and cell growth. Due to some external factors, alteration of these genes can cause fatal cancer with severity, along with the effect of the mutations. Currently, next-generation sequencing (NGS) is an emerging applied science for ascertaining the chronology of RNA or DNA to study the genetic variation correlated with cancers or other biological phenomena. It enables rapid identification of common and rare genetic variation with genome sequencing, investigation, and identification. Table 1 shows some essential genotypes and their information, such as their function within the body and their alteration effect for different types of cancer.

### 3.3. Overall Flow of Radiogenomics

So far, we have discussed radiomics and genomics separately, and their functionality. The workflow (Figure 8) of the radiogenomics (“Radiomics” + “Genomics”) study can be broadly partitioned into five different stages: image acquisition followed by image preprocessing, appropriate feature extraction, and dimensionality reduction (selection) such as PCA [38], the association of radiomics and genomics features, data analysis, and finally, the outcomes of radiogenomics. All prominent cancer types of various anatomical sections such as the brain, lungs, breast, kidney, liver, prostate, bladder, colorectal, gastric, pancreatic, ovarian, head and neck, and retinoblastoma can be studied through radiogenomics.

Different Stages of Radiogenomics:

*(i) Data acquisition and preprocessing:* Image acquisition in cancer patients is a tedious task due to the severity of the patient’s condition [14]. However, there are several medical imaging modalities, such as CT Scan, MRI, PET, Ultrasound [44,45], etc., which can locate and visualize cancer [46,47]. Each modality has its own peculiarities while considering the tissue-level radiography of various anatomical sections. The corresponding genomic data of cancer patients are collected as part of the genomics study. The preprocessing steps are an integral part while handling medical images. Preprocessing steps basically involve bias field correction, normalization, pixel or voxel resampling, and image registration [48,49]. However, data handling such as class imbalance, data augmentation, randomization, and standardization is important for cancerous images’ radiomics data [47]. The initial stage of the radiogenomics study needs the region of interest (ROI) of the radiomics data, where the exact radiomics features of cancer are available. ROI is, however, a crucial part because of the unclear margin, shape, size, and location of the tumor. The preprocessing, data handling, and segmentation of radiomics data provide better accuracy on the AI model for better diagnosis and prognosis of cancer [8].

*(ii) Feature extraction and selection:* The radiomics features in a clinical context include the essential geometric features such as the shape and size of cancer; texture features such as first-order, second-order, and higher-order texture features; intensity features of pixel and voxel values; and statistical features such as histogram analysis and wavelet features [50]. There are two prominent categories of feature-extraction process for phenotype information of cancer radiography, namely hand-crafted feature and deep features, as shown in Figure 7. Feature selection or dimensionality reduction are crucial steps for radiomics data as they lead to high dimension, which subsequently lowers the performance of the AI model. Like the phenotype information, radiogenomics study also combines the genotype information to be extracted corresponding to the phenotype information of each cancer patient. The various genotypes whose alterations can cause cancer are IDH, TP53, MGMT, EGFR, PTEN, HER2, and Ki-67, as shown in Table 1.

*(iii) Association of radiomics and genomics:* In this step, both the radiomics features and genomics features of the cancer patients are combined to understand the tissue-level characterization of the cancerous regions or non-cancerous regions from the radiomics feature [51,52,53].

*(iv) Data analysis:* This is the most important stage to demonstrate different capabilities of AI in radiogenomics, which involves various techniques such as machine learning, deep learning, statistical tools for feature extraction, dimensionality reduction, classification, prediction, cross-validation, and lesion (ROI) localization. The customary machine-learning models include the k-nearest neighbors, artificial neural network, support vector machine, logistic regression, decision tree, random forest, naive Bayes, XGBoost, and Ensemble models [54]. Several such applications of ML have been demonstrated in medical imaging such as point-based models, including problems related to gene classification and analysis [33], neonatal and infant mortality [55], and diabetes [56,57]. Further, when using point-based ML system design for classification, there have been innovations in medical imaging such as lung segmentation [58], carotid plaque classification [26,51,59,60,61], rheumatoid arthritis [62], thyroid cancer classification [63], liver [64,65], plaque tissue classification (PTC) [66], coronary [67], ovarian cancer classification [68,69,70], breast cancer classification [71], prostate cancer classification [72], skin cancer [73,74], Wilson disease [75], and ophthalmology [32]. Even though these are ML-based classification techniques, they used tissue properties from images: so-called “radiomics”. The other name is commonly known as “tissue characterization” [76].

The deep model includes the deep neural network (DNN), convolutional neural network (CNN) [77], and deep temporal models such as the recurrent neural network (RNN) and long-short term memory (LSTM) model for temporal genetic data [54]. Similarly, various statistical tests, performance evaluation parameters, and performance analysis metrics have been involved in the data analysis of radiogenomics. The lesion localization analysis can be conducted by heatmap analysis for deep diagnosis [52].

*(v) Radiogenomics outcome:* This step of radiogenomics is the decision support system that includes various endpoint outcomes such as tumor grading, survival prediction of patients, generating imaging biomarkers, clinical decision, precision medicine, risk stratification, and personalized treatment-planning of cancer patients.

**Table 1 cancers-14-02860-t001:** Some essential genotypes with their function and mutation effect.

SN	Genotype	Function	Mutation Effect	Prominent Cancers
1	TP53 (p53) [46]	Tumor suppressor gene, Initiating apoptosis, DNA repair	Genetic instability, reduced apoptosis, angiogenesis	Breast, brain, bone, leukemia, lung
2	IDH1, IDH2 [78]	Control citric acid cycle	Loss of normal enzymatic function	Leukemia, bone, brain, prostate
4	MGMT [79]	Coding for a protein that repairs DNA	Reduces binding of transcription factors and decreases gene expression; cause of glioblastomas	Brain
5	EGFR and PTEN [80]	Protein on cells helps them grow	Tumorigenesis of glioblastoma; predictor of poor survival	Brain, lung
6	ER/PR [81]	Transcription of millions of genes leads to cell proliferation	Mammary gland development and cell proliferation	Breast
7	RB1 [82]	Tumor suppressor	Blocks cell-cycle progression	Retina, brain
8	Histone H3 [83]	DNA repair	Poor prognosis	Brain, bone
9	ATRX [84]	Protein formation for normal development	Intellectual disability, genital abnormalities, hypotonia, facial disorder	Brain
10	BRAF [85]	Encode B-Raf protein	Melanoma and colorectal cancer	Skin, colon
11	HER2 [86]	Control cell growth	Breast, bladder, ovarian, pancreatic, and stomach cancers	Breast, ovarian, pancreas, lung
12	Ki-67 [87]	Cell proliferation, Prevent aggregation of mitotic chromosomes	Inhibition of ribosomal RNA; synthesis; prostate, brain, and breast carcinomas, nephroblastoma, and neuroendocrine tumors	Prostate, brain, breast, kidney
13	PD-L1 IHC [88]	Controls the induction and maintenance of immune tolerance within the tumor microenvironment	Squamous cell carcinoma	Skin
14	NF1 [89]	Production of neurofibromin protein for cell growth	Deprivation of neurofibromin only causes uncontrolled cell growth	Skin, nervous system
15	MYB family [90]	Proliferation and differentiation of hematopoietic progenitor cells	Deletion in the C-terminal domain that causes cancer	Leukemia, glioma
16	BRCA1 and BRCA2 [91]	Repair damaged DNA and tumor suppressor	Abnormal cell growth, which can lead to cancer	Breast, ovarian
17	CDKN2A/B Family [92]	Produce the p14(ARF) and p16(INK4A) proteins.	40% of melanoma, 95% of pancreatic tumors	Melanoma, glioblastoma, and pancreatic
18	MSH2, MSH6, and MLH1 [93]	Repair damaged DNA and tumor suppressor	Lynch syndrome; complete loss of MSH6 protein production.	Colon
19	CDH1 [94]	Produce protein called epithelial cadherin or E-cadherin	Hereditary diffuse *gastric cancer* (HDGC)	Gastric
20	KRAS [95]	Making a protein called K-Ras	32% of lung cancers; 85% to 90% of pancreatic cancer; 40% of colorectal cancers,	Lung, pancreatic, colorectal
21	PBRM1 [96]	Tumor suppressor, chromatin remodeling	40% of clear cell renal cell carcinoma (ccRCC)	Kidney
22	TERT [96]	Produce enzyme called telomerase; protect from chromosome degrading	Potential as biomarkers of various cancer	Brain, melanoma, leukemia
23	SMARCB1 [97]	Chromatin remodeling	Coffin–Siris syndrome (CSS)	Brain and kidney
24	*PDGFRA* [98]	Produce protein PDGFRA	Amino acid residue changes	Gastric

## 4. The Era of Radiogenomics in Precision Medicine

The popularity of precision medicine has grown over the last few decades, especially in oncology care. Precision medicine involves optimizing medication according to an individual’s phenotypic and genotypic characteristics and the nature of the disease, taking the ‘one size fits one’ approach [99,100]. This encompasses mathematical modeling and biology, including metabolomics, transcriptomics, proteomics, and genomics [99]. Precision medicine involves identifying specific treatment targets and developing means of checking the changes in these targets using non-invasive and reliable methods [99]. Artificial intelligence in ‘-omics’ and imaging modalities have been utilized in this context to develop models that can predict changes in the targets’ environment and monitor the therapeutic outcomes, apart from the available standard care [101].

The shift from the traditional ‘one size fits all’ to the ‘one size fits one’ route involves implementing advances across several cross-sectoral, interdisciplinary, and multidisciplinary fields [102]. These advances range from developing tools for big data analysis and research in individualized medicine to standardization of possession, repositioning, and sharing of patients’ computerized health reports and the involvement of the patients themselves [103,104]. The key to the success of precision medicine is a computationally intensive task. To develop computationally effective means of merging radiomics, genomics, and clinical data for data mining [102], the use of radiogenomics research demonstrates its significant potential for developing non-invasive diagnostic and prognostic markers, especially in the field of oncology [102].

Over the past few years, the rise of radiogenomics in cancer medicine can be attributed to various factors [105]. First is the gap existing between molecular pathology and traditional radiology. A deeper understanding of tumor components has driven the development of tailored cancer therapeutics [106,107]. Second is the rising interest in incorporating artificial intelligence with oncology medicine. The application of ML algorithms to a large-scale imaging database has further driven this process [108]. The third is the growing understanding of the tremendous potential that imaging data hold. This is of particular importance in the case of oncology, where there are temporal and spatial limitations of tissue sampling [105].

As discussed, radiomics involves extracting quantifiable data available from clinical radiography and integrating these data with patient data to generate a searchable database. Radiogenomics is then utilized to provide complete information for a heterogeneous tumor or a metastatic disease, guiding the development of therapy suitable for the individual [99]. Large-scale databases are possible due to the enormous amount of imaging data. Mining important and significant radiomics information from these radiological databases is necessary to generate valuable information employing advanced techniques, frameworks, analytics, and algorithms [100]. However, the more significant challenge remains in maintaining clarity and consistency in performing such studies. Hence, developing a standardized workflow and internationally consented methods is necessary for effective and robust studies [100].

Consortiums have been developed to standardize radiogenomics studies [100]. The Transparent Reporting of a multivariable prediction model for individual prognosis or Diagnosis (TRIPOD) report is a set of instructions prepared to cover the studies with validation or development of different multivariable prediction models [109]. The Image Biomarker Standardization Initiative (IBSI) was founded to deliver a standard for calculating commonly used radiomics features (machine and deep) and the image-processing technique required earlier for radiomics features [100,110]. For clinical translation, standardization of methods of radiomics is a prerequisite. Apart from standardizing, detailing of quality of the radiomics study is also of utmost importance. Radiomics researchers should practice findability, accessibility, interoperability, and reusability (FAIR) ushering philosophy and ensure that the research objectives are findable, interoperable, accessible, and recyclable [111]. This will ensure validation and quality assurance of radiomics study.

## 5. What Have We Achieved So Far in Radiogenomics?

Radiogenomics can prove to be a beneficial tool for optimal patient selection in oncology [101]. It can act as a digital, non-invasive biopsy technique with the ability to identify and quantify tumor infiltration and helps with the development of personalized immunotherapy regimens and continuous monitoring of therapeutic response. The combination of imaging data with radiomics looks promising to improve disease diagnosis, prognosis, and the prediction of disease outcomes [101]. This field’s progress can be well illustrated with the application of radiogenomics studies in numerous cancers such as glioblastoma, hepatocellular carcinoma, non-small cell lung cancer, hematopoietic tumors, etc. [101,102].

The dawn of radiogenomics imitates an alteration in research from the radiology–pathology level to a genetic level [100]. Over the past decade, radiogenomics has experienced steady growth by mining radiomics, genetic, and clinical data [102]. The development of deep learning and big data programming is instrumental in radiogenomics research and contributes to the development of newer algorithms, workflow, and methods [112]. A significant achievement in radiogenomics is developing a fully automated system combined with a radiological workflow, shown in Figure 9 [113]. This reduces the overall time involved in performing repetitive and tedious tasks while improving efficiency and productivity [114,115]. Another advantage is the real-time monitoring of treatment by simultaneously comparing several images from the database [113,114].

The success of radiogenomics in developing personalized medication regimens is highly dependent on the reproducibility and transparency of the predictive tools and programming algorithms [116]. The availability of guidelines such as the TRIPOD has played a crucial role in progressing towards these goals [100]. At the same time, it is of utmost importance to see that applying these advanced radiogenomics methods accounts for the intricacies of the in-place radiobiology knowledge [116]. Imperfect and faulty datasets available in a radiogenomics database may be conjugated with prior knowledge of the outcomes to establish new conclusions [117].

**Figure 9 cancers-14-02860-f009:**
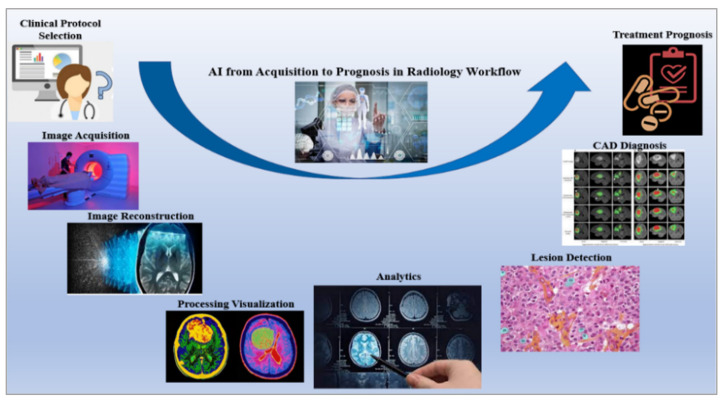
AI improves entire radiology workflow from clinical protocol selection to the treatment prognosis [118].

## 6. Artificial Intelligence in Radiogenomics

### 6.1. What AI Offers: A Computational Perspective

Advancements of different estimation techniques, such as genomic sequencing and medical radiography of cancer, have enormously augmented the quantity of patient data accessible to the clinician perspective to radiogenomics. Indeed, AI, the advanced set of computational algorithms, is perfect for this and can easily deal in radiology from image acquisition, image reconstruction, feature extraction and selection, data analysis, and developing models to analyze cancer, treatment prognosis, follow-up planning, and many other aspects [8]. Figure 10 represents how AI improved the entire radiological workflow in current clinical practice.

The reason for choosing AI in radiology (radiogenomics) is its excellent handling of a considerable proportion of data compared with the traditional statistics-based methods. AI-trained models recognize the data by analyzing patterns using different phenotypes and (or) genotype features. Further, these models can be used in predicting (estimating) unseen cohorts to check and validate the accuracy. Apart from classification or regression techniques in radiogenomics, AI could be used in several other applications such as cancer heterogeneity analysis, progression of tumors, recurrence, etc. [8]. AI improves the entire radiological workflow in three key ways: productivity, quantity, and precision. Productivity increases via automation and prioritizing routine jobs. In terms of quantity, it can extract and quantify information semi-automatically or fully automatically. For precision, by ensuring that the correct information is accessible, this is obtained by separating unnecessary information. AI, machine learning, and deep learning are interchangeable terms and create some confusion. Basically, AI provides broad ways of designing intelligent methods through radiological data mining that can efficiently and creatively address radiological problems. Under the umbrella of AI, numerous ML algorithms such as artificial neural networks, support vector machines, decision trees, random forest, and k-nearest neighbors have proven phenomenal. Again, neural networks have been working as parental concepts that range from very simpler to complex architectures, such as multilayer perceptron (MLP) and deep learning (DL). The following diagram depicts AI and its subsets in the prospect of radiological data analysis. The Venn diagram in Figure 11 presents the role of AI in its components in oncology care perspective to different applications in imaging and digital pathology, drug discovery, precision oncology, patient data management system, next-generation sequencing, etc.

Machine learning, a subspace of AI that maneuvers the medical imaging features added with genomics features such as mutational status, could be used in several applications such as classification, regression, clustering, dimensionality reduction, and density estimation presented in the following Figure 12. These methods can be classified based on the type of learning it provides, such as supervised, unsupervised, and reinforcement learning; in the first one, which is most frequent in radiogenomics, the data are labeled prior to the training procedure. Further, these labels are also used as the reference standard to assess algorithm performance in the test cohort [119]. No prior labels are considered for unsupervised learning, and the algorithms/methods automatically cluster the given inputs based on specific characteristics [120]. In reinforcement learning, the algorithms/methods learn based on the non-stop response to their performance in the particular assigned task, or we can say that from its errors [120]. To conclude, all these methods may be united to augment forecast performance and analysis in the radiogenomics study of cancer, as shown in the following figure. Recent radiogenomics studies based on machine learning produced encouraging outcomes for cancer prognosis and treatment planning [8]. However, it is clearly observed the trend is shifting from machine learning to the end-to-end deep-learning model [121]. Deep learning, a subset of machine learning, consists of algorithms motivated by artificial neural networks. A neural network contains several layers with a number of nodes. First, there is an input radiological image or lesion (tumor or certain ROI), followed by several hidden layers. In the end, the output layer comprises the queries the network is assumed to respond such as tumor type classification, survival prediction, etc. The basic flow is given below in Figure 13.

Basically, a node at each layer, the output of the previous layers gets computed and further passed to the next layer. Specifically, training the deep neural network is nothing but figuring out the best output for an individual node, and when all the nodes are united, the deep model produces the correct response or outcomes. Radiogenomics studies [24,122,123,124] based on deep learning produced a significant outcome. Therefore, it has been seen that AI including machine- and deep-learning perspectives to radiogenomics study of cancer plays a very significant role.

### 6.2. Cross-Validation: A Crucial Step

Cross-validation is a technique to measure the effectiveness of an AI-based model. In radiology, it has a very crucial role to make a generalizable model. Basically, it is a resampling technique that evaluates how a model will perform on the independent cohort. If an algorithm is not properly cross-validated then it gives a very biased results perspective of accuracy. In Table 2, a summary of the different types of cross-validation is presented.

### 6.3. Performance Metrics: An Essential Step in the Evaluation of the AI Models

Accurate evaluation of the algorithm is a very important step. Table 3 includes all the metrics used for evaluation in different studies of radiogenomics in oncology.

### 6.4. Is AI Efficient for Radiogenomics?

The application of AI in radiogenomics is highly promising [125]. The first step involved using AI at the detector level to process data for reconstructing images, which also includes corrections for scattering, attenuation, etc. [126]. Further use of AI includes processing images (fusion [48], segmentation [74], etc.). Finally, AI is used to generate models for personalized medicine based on information extracted from the images obtained from the database [126]. The quality, generalizability, and robustness of the algorithms/classification models will determine the radiomic and artificial intelligence [127].

Although AI and radiogenomics research is growing exponentially, clinical implementation is yet to be achieved. For improved implementation, models should be presented so that clinicians can understand results and interpret them adequately for carrying out appropriate treatment decisions. Models must be validated and well trained and, at the same time, must be transparent in providing risk information as per an individual’s prediction [127].

## 7. AI in Radiogenomics Studies of Different Cancer Types

According to the World Health Organization (WHO) [128], brain, breast, and lung cancer, etc., is one of the leading causes of death in the global population with an average age of 70. In Table 4, the role of AI-based radiogenomics in different types of cancer studies is explained briefly in terms of the motivation of the research, radiomics and genomics information, AI-based imaging signatures for predicting the status of genomics, cohort information of radiogenomics, and different performance metrics involved with limitations and suggestions.

It is observed that both machine- and deep-learning imaging signatures have been used in nearly the same proportions for predicting the status of genetics information from the radiomics features provided. However, ML-based signatures have been used more, as discussed in statistical analysis in Section 2 along with Figure 3a. The application and cohort sizes play a premier role for the selection of AI models using machine-learning or deep-learning paradigms [129,130]. For supervised learning using smaller cohort sizes, augmentation is adapted during the training framework, while no augmentation is applied for the testing data. The ML-based imaging signatures provide the hand-crafted radiomics features based on shape, size, grade, tissue, texture, histogram, etc., whereas for those that are DL based, we have generated an automatic deep radiomics feature to predict the model behaviors for genetics information. Therefore, the DL-based imaging signature provides more automation power in comparison to the machine-learning models. As a result, the researchers are biased toward ML-based imaging signatures when they want to avoid more training time; also they have a lower amount of data with which to train the models. Additionally, deep learning is a data-hungry model that needs more radiogenomics data with more training time to provide more precisive results. The most fundamental challenge in the current research is the hands-on access to the radiogenomics datasets. It is recommended that such datasets be publicly available for the development of the advanced AI tools leading to product design for clinical settings. Hence, there is a trade-off between both technologies for predicting the status of genomics information in case of certain tumor. Under the machine-learning methodologies, the SVM, RF, DT, NB, XGBoost, Ensemble, Univariate, and multivariate regression models are used in more proportion compared to other ML or DL models.

The radiogenomics approach of diagnosis using artificial intelligence is gaining increased popularity in the case of more frequently occurring cancers such as breast, brain, and lung cancers. The frequently used genomics prediction using AI-paradigm for the cancer types, namely MGMT, IDH1/2, BRCA1/2, Lumina A/B, ER, PR, EGFR, Ki-67, and HER2, are shown in the table below. Again, due to a lack of sufficient genomics data availability, machine-learning approaches are preferred for these types of tumors. Since radiogenomics is more prevalent in medical sciences compared to engineering sciences, to avoid a larger delay in learning curve by the medical practitioners, AI tools such as machine learning and deep learning are likely to prove a foundational strategy for patient management [131,132]. The performance metrics generated by the AI-based imaging signatures such as accuracy and area under the curve (AUC) are mostly adopted by many authors as the standard parameters. However, other parameters such as sensitivity, specificity, precision, F1-score, and other statistical measurement tools can be used for further validation of the performances of the AI models.

By going through Table 4 (a–h) we can clearly observe the significant role of AI radiogenomics in different types of cancer in the era of precision medicine. Table 4 (a) discusses some recent AI-based studies in radiogenomics for breast oncology care. This cluster of studies basically focuses on the genomics status prediction of the most prominent genetic mutant in case of breast cancer such as BRCA1/2, Luminal A/B, HRE2, Ki67, ER, and PR. To predict the status of these relevant mutants, machine-learning models such as SVM, RF, DT, NB, XGBoost, Ensemble, Univariate, and multivariate regression models seem to be very promising, as of adaptation by several researchers. However, deep-learning models have been used in some scenarios. To implement the radiogenomics aspects in case of breast cancer care involves a few popular datasets such as The Cancer Imaging Archive (TCIA), The Cancer Genome Atlas (*TCGA*), and Full-Field Digital Mammography (*FFDM*). The standard performance measure metric used to check the imaging signature is area under the curve (AUC). As per the studies considered, the typical performance of these models for the above said genomics is near ~60% and ~80% in all cases.

Table 4 (b) focuses on brain oncology care using AI-based imaging signatures. Some more promising studies predicting the brain cancer mutant are considered here. The frequently used mutants are IDH1, IDH2, MGMT, EGFR, PTEN, PDGFRA, CDKN2A, TP53, and RB1. It can be concluded from the considered radiogenomics studies that the most common type of genetic alterations of brain cancer are IDH1, IDH2, and MGMT. To predict the status of these relevant types of genomics, the relevant imaging signatures found are a mixture of both machine and deep learning. However, ML-based imaging signatures such as logistic regression, XGBoost, random forest, and decision tree are predominantly used compared to DL-based models. The common cohort available for the radiogenomics of brain cancer using AI-paradigms are The Cancer Imaging Archive (TCIA), The Cancer Genome Atlas (*TCGA*), and some personalized data from multicenter hospital environments. The performance metrics for measuring the model performances are under the curve (AUC) along with accuracy, sensitivity, specificity, precision, and f1-score. The standard AUC for these studies is near ~85%. The accuracy parameters values are near ~85%.

Lung cancer is the most observed cancer type and has been discussed in Table 4 (c). The genetic mutation of genes such as EGFR, KRAS, TP53, and a few RNA sequencings is considered to be the most frequently associated. To predict the genomics status, the most popular AI-based imaging signatures include a mixture of both ML and DL. This includes models such as CNN, 3DCNN, SVM, random forest, and generalized ML-linear models. The common type of database for the radiogenomics of lung cancer include The Cancer Imaging Archives (TCIA) and other multi-institutional databases. The AUC is the most observed performance metric with a mean value of ~80% from the cluster of models considered under this category.

Further, Table 4 (d) discusses some recent AI-based models in radiogenomics for liver oncology care. The motivation for this category of cancer diagnosis using radiogenomics is the prediction of early recurrence of Hepatocellular carcinoma (HCC) and the survival prediction involved in it. The genomics alteration for the HCC includes TP53, TOP2A, CTNNB1, CDKN2A, AKT1, alpha-fetoprotein, AFP, and DCP. Again, the machine learning imaging signature is considered to be the preferred one, with the standard mean AUC observed as ~85%. However, other statistical measures of Kaplan–Meier analysis and Cox regression are also involved in the measurement process.

The recent AI-based models in radiogenomics for prostate oncology care have been discussed in Table 4 (e). The common type of diagnosis for prostate cancer involves the prediction of tumor aggressiveness in the prostate. Machine- and deep-learning models are used for this purpose with standard performance metrics AUC and accuracy. In these studies, authors focused on deep-learning-based models such as CNN, Resnet 101, and LSTM for building radiogenomics-based models and obtained very promising AUC that is more than 0.9. However, to make more generalizable and robust models, the use of multi-institutional cohort is highly recommended for future prospects.

Table 4 (f) discusses the involvement of an AI imaging signature for studies in radiogenomics for ovarian oncology care. The major task under this category of care involves the prediction of PFS in advanced HGSOC and PM in ovarian cancer. Here, mainly machine-learning models such as KNN, SVM, Logistic regression, and ensemble-based learning have been taken into consideration for building models with very promising AUCs around 85%. From a database point of view, even though The Cancer Imaging Archive (TCIA) is the common database, multi-institutional databases are also used for this purpose. For future prospects, the use of multiple multiparametric scans is highly recommended instead of using only single modality for ovarian-based radiogenomics studies.

Table 4 (g) discusses the oncology studies of collateral with radiogenomics using AI imaging signatures. Prediction of KRAS mutations is the measure of genomics associated with this type of cancer. The other mutants involved here are NRAS, BRAF, and AF. To determine genomic mutational status, machine-learning models such as SVM, Naïve bayes classifier, decision tree, and RELIEF have been implemented. However, they obtained AUC in the majority of the studies, which is more than 85%. Various public medical institute databases and TCIA have been used for building predictive models. An increase in the testing cohort size is highly recommended here to make a robust imaging (genomic) signature.

Table 4 (h) narrates some recent AI-based imaging signatures in radiogenomics of gastric oncology care. Predicting lymph node metastasis and prediction of PD-L1, and PM status in gastric cancer (GC) is the major objective under radiogenomics using AI. The machine-learning paradigms such as SVM, decision tree, random forest, and multivariate logistic regression are dominating in predicting the status of genomics over deep-learning models. Traditional machine-learning features such as intensity, first- and second-order statistics are very promising to analyze imaging phenotypes in gastric cancer with impressive AUC more than 75% for the testing cohort.

Though radiogenomics studies of cancer using AI-paradigms have several key benefits, there are certain challenges (described in the next section) that demonstrate why AI in radiogenomics is a bit concerning for oncologists to use frequently in current clinical practice.

**Table 4 cancers-14-02860-t004:** Recent AI-based studies in radiogenomics for various oncology care.

CIT *	Motivation	Radiomics Information	Genomics Information	AI-Based Models	Dataset	PM ^$^	Performance Measure	Outcomes
**(a)**								
[133]	Risk assessment in breast cancer	Traditional Radiographic, Texture Analysis, Pretrained CNN for deep features	BRCA1/2	SVM Model	456 clinical FFDM patients	AUC	BRCA1/2 gene-mutation: AUC = 0.86 unilateral cancer patients: AUC = 0.84	Fusion classifiers performed significantly better. Deep features performed very well.
[134]	Association Assessment of imaging phenotype with molecular subtype.	529 tumor and tissue imaging features.	Luminal A, ER, PR, EGFR, Ki67, HRE2	ML-based multivariate models	922 patients (Proprietary data)	AUC	Luminal A subtype: AUC = 0.697, TNBC: AUC = 0.654, ER: AUC = 0.649%, PR: AUC = 0.622%	Application in early diagnosis with association relation between the MRI-based imaging features and genomic features.
[135]	Prediction of molecular subtypes and prognostic biomarkers	CT perfusion features include lymph node status, tumor grading, tumor size	ER, PR, Luminal A, Ki67, HRE2	SVM, RF, Decision tree, Naïve Bayes	723 patients (Proprietary data)	AUC, ACC.	Random Forest: AUC: 0.86, Tumor grade and size: AUC: 0.88 and 0.85 ER and PR status: AUC: 0.88 and 0.85 HER2 and Ki67: AUC: 0.88 and 0.85 Molecular subtypes: AUC: 0.82	Helps in non-invasive diagnosis by performing a depth analysis of the relation between molecular subtype and CT-based imaging features.
[123]	Classification of breast cancer molecular subtype	Deep features	Luminal A	Google Net, VGG, & CIFAR network	272 patients (Proprietary data)	AUC	Deep features: AUC = 65% TL: AUC = 60%	Provides a non-invasive way to detect Luminal A tumor subtype with the help of DL.
[136]	Diagnosis of breast cancer	Features: tumor shape, size, morphology, enhancement texture, enhancement-variance kinetics, and kinetic curve assessment.	RNA sequencing, KEGG, GSEA	Radiogenomics	TCGA/TCIA	-	-	Detailed analysis of the association between the gene pathways and imaging features provides a future direction for the non-invasive diagnosis of breast cancer.
**(b)**								
[137]	CAD system	Traditional features: morphological, intensity, and textural features.	IDH1	Logistic regression	32 (WT IDH) and 7 (mutant IDH) patients from TCIA	ACC, SENS, SPEC	Morphology: ACC = 51% (20/39), SPEC = 50% (16/32), SENS = 57% (4/7); Intensity: ACC = 59% (23/39), SPEC = 59% (19/32), SENS = 57% (4/7); Texture: ACC = 85% (33/39), SENS = 86% (6/7), SPEC = 84% (27/32).	Non-invasive diagnosis of tumor CAD system.
[138]	Prediction of IDH1 for LGG tumor	Texture, intensity, shape, and wavelet features.	IDH1	CNN	151 patients from the Department of Neurosurgery, Huashan Hospital.	AUC, ACC, SPEC, SENS, NPV, PPV, MCC	IDH1 estimation, in radiomics method: AUC = 86%, DLR: AUC = 92%, DLR based on multiple-modality MRI, AUC = 95%	Provides a direction for early researchers to choose the models as it gives a comparative performance analysis of DL-based radiomics and normal radiomics methods.
[139]	Classification of MGMT promoter	Nine textures, histogram, gray level-based features	MGMT, IDH1	XGBoost	262 subjects from TCGA and TCIA	AUC, ACC, SENS, SPEC, F1 score	AUC = 89.6%	Yields better treatment planning for patients with IDH1 wildtype GBM in the primary diagnosis phase.
[140]	Characterization of genetic heterogeneity over enhancing and non-enhancing tumor.	MR imaging texture features	EGFR, PTEN, PDGFRA, CDKN2A, TP53 and RB1.	Predictive decision-tree models.	18 GBM Patients (Proprietary data)	ACC, LOOCV,	Accuracy for 6 driver genes: EGFR = 75%, PDGFRA = 77.1%, CDKN2A = 87.5%, TP53 = 37.5%, RB1 = 87.5%,	In primary diagnosis and better treatment planning of patient with GBM.
**(c)**								
[141]	Prediction of EGFR and KRAS mutation	Texture and Non-texture features	EGFR and KRAS	Ensemble model based on ML and CNN.	99 patients from the TCIA	AUC, ACC, SENS, SPEC	AUC for ML models: EGFR = 75%, KRAS = 72%, For DL models: EGFR = 82.8% KRAS = 72.2%.	Enhancing the performance of non-invasive diagnosis of lung cancer by predicting EGFR and KRAS mutation in a small dataset
[142]	Prediction histology and tumor Recurrence.	117 radiomic features based on GLM.	KRAS, TP53, EGFR	ML and Generalized linear model	151 Institutional databases	ACC, F1-score	AUC = 87%	Compressive analysis of showing a correlation between genomic and tumor subtype.
[143]	Prediction of tumor Recurrence in Non-small cell lung cancer (NSCLC.	Handcrafted: GLCM, histogram-based statistics, Laplace of Gaussian. Deep features	The RNA-sequencing.	Genotype-guided radiomics method	162 patients from the TCIA dataset	AUC, ACC, SENS, SPEC	AUC = 76.67% and ACC = 83.28%	Showing an effective prediction method with low cost and improved accuracy.
[144]	Risk prediction of lung cancer	Feature: patient’s current and prior CT volumes	-	3D CNN	6716 National Lung Cancer Screening Trial cases	AUC	AUC = 94.4% in risk prediction	Clinical validation proves its low-biased performance and allows enhancement of the screening process via CAD and automated screening to the radiologist.
[143]	Classification of histology subtype	1695 quantitative radiomic features (LOG, GLCM)	Histological subtypes	Incremental Forward Search and SVM	278 patients (181 NSCLC and 97 SCLC)	AUC	SCLC vs. NSCLC: 74.1%, SCLC vs. AD: 82.2%, SCLC vs. SCC 66.5% and AD vs. SCC: 66.5%	Detailed analysis of phenotypic variation exists among various lung cancer histological subtypes in CT images.
[145]	Classify somatic mutations	Radiomic signature including tumor volume and maximum diameter, intensity.	EGFR and KRAS	Random Forest	Four independent datasets (PROFILE, TIANJIN, MOFFITT, xHARVARD-RT)	AUC	AUC: 80% EGFR+ and KRAS+, 69% with EGFR+ and EGFR−, 63% with KRAS+/KRAS− radiomic signatures	Relation between the imaging phenotype captured with a genotype and EGFR mutant tumors has a clinical impact in selecting patients for targeted therapies.
**(d)**								
[146]	Prediction of early recurrence of HCC	21 CT image-based radiomic signature	-	Machine learning	Proprietary data (215 HCC patients)	AUC, SENS, SPEC	Radiomic features: AUC = 81.7%, clinical data AUC = 78.1%, combined model AUC = 83.6%	Shows a direction towards preoperative estimation in early prediction of recurrence less than 1 year and helps radiologists with better treatment planning.
[147]	Diagnosis in HCC	Features include texture features, first-order histogram, and GLCM.	TP53, TOP2A, CTNNB1, CDKN2A and AKT1	Machine learning	27 patients from TCGA, and TCIA.	AUC, SPEC, SENS.	TP53: AUC = 86.61%, TOP2A 78.0%, CTNNB1: 86.8%	Ability to categorize HCC tumors on a genetic level which helps the radiologist for early diagnosis of HCC patient
[148]	Prediction of progression-free survival (PFS) and overall survival in uHCC	SUV statistics, co-occurrence matrix, neighborhood intensity, neighborhood gray level dependence	Alpha-fetoprotein	Machine learning	Proprietary data (371 patients)	-	For survival PFS: [PFS-pPET-RadScore < 0.09] vs. 4.0 mo [95% CI(Confidence Interval): 2.3–5.7 mo] in high-risk group. median of 11.4 mo [95% CI: 6.3–16.5 mo] in a low-risk group. [OS-pPET-RadScore < 0.11] vs. 7.7 mo [95% CI: 6.0–9.5 mo] in high-risk group.[PFS-pPET-RadScore > 0.09]; *p* = 0.0004) and OS(Overall Survival): median of 20.3 mo [95% CI: 5.7–35 mo] in low-risk group. [OS-pPET-RadScore > 0.11]; *p* = 0.007)	Helps in better treatment planning for the patients undergoing transarterial radioembolization using Yttrium-90.
[149]	Prediction of overall survival in HCC	Features including maximum diameter, histogram-based texture features	AFP, DCP	Machine learning	178 patients (Proprietary data	Kaplan-Meier analysis	Random survival forest model’s high and low predicted individual risks are *p* = 1.1 × 10^−4^ for DFS, 4.8 × 10^−7^ for OS respectively, and based on the multivariate Cox proportional hazards model, high predicted individual risk (hazard ratio = 1.06 per 1% increase, *p* = 8.4 × 10^−8^)	OS prediction shows a better direction towards the improving survival of the patient.
**(e)**								
[150]	Diagnosis of prostate cancer	Features: Gabor texture, Gleason grade, and gland lumen shape	Gleason score, QH	ML	54 patients from UPenn and 17 patients from SV	AUC	Prediction of Gleason grade based on Gabor texture features AUC = 69%, prediction of QH based on gland lumen shape features AUC = 0.75	Relation between in vivo T2w MRI phenotype predicting prostate cancer status.
[151]	Prediction of tumor aggressiveness in prostate	Multiparametric (mp) MRI and 68Ga-PSMA-PET/CT phenotypes.	CNAs	-	5 patients of the University of Heidelberg	-	Highly significant CNAs (≥10 Mbp) were found in 22 of 46 biopsies.	Correlating the most aggressive lesion with imaging features helps in future prostate cancer diagnosis and prognosis.
[152]	Diagnosis of prostate cancer	Texture Based features, morphological features	-	LSTM and ResNet101	230 for MRI by the Health Insurance Portability.	AUC, SENS ACC, SPEC, NPV, PPV, MCC	LSTM: AUC = 0.9999, ResNet − 101AUC = 100%	Detection of prostate cancer prediction is better on a DL-based model.
[153]	CAD for prostate cancer	564 radiomic features of texture, intensity, shape, and orientation.	-	CNN DL, radiomic model.	644 patients from healthcare centers in Netherland.	AUC, ACC, SENS, SPEC.	DL: AUC = 89%, Active Surveillance dataset using Radiomic model AUC = 83%	Developed a tool for significant-PCA classification with radiomic model.
**(f)**								
[154]	predicting early recurrence in HGSOC	Radiomic nomogram	-	KNN, SVM, and LR	Proprietary data (256 patients)	AUC, Kaplan-Meier survival analysis and Decision curve analysis	C-index for clinical factors model = 82% [95% CI (0.75–0.88)] (training set) (validation set): 77% [95% CI (0.59–0.90)] Radiomics nomogram C-index = 0.91 [95% CI (0.85–0.95)] (training set),, the C-index = 0.85 [95% CI (0.69–0.95)] (validation set)	Helps in early individualized recurrence prediction in patients with HGSOC
[155]	Classification of ovarian cancers (SOCs).	Features include Histogram, Formfactor, GLSZM, RLM.	CEA, CA125	ML	Proprietary data (110 patients)	AUC, SPEC, SENS	AUC = 85.4%	The model using radiomic features of arterial phase of CT with clinical features is the first study to develop a useful tool for differentiating the POC and SOC.
[156]	Prediction of PM in ovarian cancer.	Radiomics features: T2WIs, T2WIs, multi-value DWIs	-	LR	89 patients Shanxi Medical University	AUC.	AUC = 96.3% (training) AUC of 0.928 (validation)	Treated as a biomarker for risk stratification.
[157]	Prediction of PFS in advanced HGSOC.	Imaging features	Pelvic fluid, and CA-125		261 patients (Proprietary data)	AUC	AUC = 96.9%	The quantitative solution to predict PM in OC patients.
[158]	Assessments of CT imaging features of HGSOC	Ovarian mass, size of pleural effusions and ascites, mesenteric implants and infiltration, lymphadenopathy, and distant metastases.	-	ML	92 patients (Proprietary data)	Estimates of Krippendorff α and coverage probabilities	Pleural effusion and Ascites: α = 0.78, Intraparenchymal splenic metastases: α = 0.08	Experimental results show evidence of the clinical and biological validity of these image features.
**(g)**								
[159]	Prediction of mutation status and prognostic values in colorectal cancer	-	PIK3CA exon 9 and 20, NRAS exon 2 and 3, KRAS exon 2, 3 and 4, and BRAF exon 15	PCR and direct sequencing	353 CRC patients at Zhongda Hospital	-	13.9% (49 out of 353) CRC patients carried mutations at RAS exons outside the KRAS exon 2.	Provides the importance of these novel molecular features in CRCs
[160]	Prediction of KRAS/NRAS/BRAF mutations in colorectal cancer (CRC).	Features include shape features, GLCM features, and GLRLM features.	KRAS/NRAS/BRAF	RELIEFF and SVM	117 patients (Proprietary data)	AUC, SENS, SPEC.	Prediction of KRAS/NRAS/BRAF mutations, AUC = 86.9%	The predicted association is useful for the analysis of tumor genotype in CRC and hence helps in therapeutic strategies.
[161]	Prediction of KRAS mutations using MRI	polypoid pattern, axial tumor length	KRAS	-	275 patients (Proprietary data)	-	The frequency of KRAS mutations was higher in the N2 stage (53.70%), and polypoid tumors (59.09%).	Helps in finding the imaging predictor of KRAS which helps the radiologist to make a better therapy strategy.
[162]	Prediction of the mutation status molecular subtype in colorectal cancer.	Features: tumor size, degree of the tumor, C-reactive protein level, differentiation, and TNM stage	KRAS	Machine Learning	58 patients (Proprietary data)	AUC	AUC on predicting the KRAS mutant = was 86.5%	Provides a higher performance for the prediction of the KRAS mutation status in CRC.
[163]	Classification of imaging predictors.	-	KRAS	Naive Bayes classifier	457 patients (Proprietary data)	-	-	Ability to identify disease course relation with mutated oncogenes and provides a cheaper, quicker substitute for genome sequencing.
**(h)**								
[164]	Predicting of lymph node metastasis.	Features include intensity features, shape, GLZLM, GLRLM, GLCM.	-	SVM	490 patients (Proprietary data).	AUC	LN+, AUC = 82.4% (training and validation), AUC = 76.4% (test data)	Shows a promising tool for the preoperative prediction of LN status in patients with GC.
[115]	Prediction of PD-L1 status in gastric cancer (GC).	-	PD-L1	SVM and RF	358 patients of Nanjing Drum Tower Hospital	AUC	Using SVM AUC = 70.4%, 79.9% in primary and validation cohort.	A promising tool to predict PD-L1 status and helps to improve clinical decision-making about immunotherapy.
[165]	PET based radiomic model for prediction of PM of gastric cancer.	Features including GLCM, GLZLM, NGLDM, and GLRLM	CA 125, PM, SUVmax.	Multivariate LR	355 patients (Proprietary data).	AUC	Radiomics model: AUC = 86%, 87%, Clinical prediction model: AUC = 76% and 69%	Provides a novel tool for predicting peritoneal metastasis of gastric cancer.
[166]	Prediction and investigation of the efficiency of neoadjuvant chemotherapy in survival stratification.	Texture, filter transformed, and wavelet features.	-	Randomized tree	106 patients (Proprietary data)	AUC	Rad_score: AUC = 82%, clinical score: AUC = 62%	Effective prediction treatment for neoadjuvant chemotherapy and stratifying patients into various survival groups.
[167]	Predict the status of lymph node metastasis (LNM).	Shape-based features, first-order based, texture-based features.	Genome stable, Epstein–Barr virus-positive, chromosomal and microsatellite instability.	Multivariate LR	768 patients (Proprietary data)	AUC	AUC = 92% (training cohort), AUC = 86% (validation cohort) AUC = 85% (EGC patients)	Serves as a non-invasive tool for preoperative evaluation of LNM in EGC.

Note: ACC—accuracy; SPEC—specificity; SENS—sensitivity; HCC—hepatocellular carcinoma; NSCLC—non-small cell lung cancer; small-cell lung cancers (SCLC); LOG—laplacian of Gaussian; HGSOC—high-grade serous ovarian cancer; CAN—chromosomal copy number alterations; CA 125—carbohydrate antigen 125; QH—quantitative histomorphometry; UPenn—University of Pennsylvania; SV—St. Vincent’s Hospital; GLZLM—gray-level zone length matrix; NGLDM—neighborhood gray-level dependence matrix; GLRLM—gray-level run-length matrix; GLCM—Gray-level co-occurrence matrix; MCC—Mathews correlation coefficient; GLSZM—gray-level size zone matrix; RLM—run-length matrix; PM—peritoneal metastasis; PES—progression-free survival; LNM—lymph node metastasis; T2-weighted images—T2WIs; fat suppressed—T2WIs; diffusion-weighted images—DWIs; Logistic Regression—LR; Machine Learning—ML; CIT *—citations; PM ^$^—Performance metrics.

## 8. Benchmarking: Comparison between Different Radiogenomics Reviews

We have observed that recent progress in AI and genomic sequencing of cancers provided new hope in radiogenomics study in individualized and precision medicine. For this reason, it has drawn significant attention from researchers and scientists. Recently, some reviews based on radiogenomics have been performed by different groups around the world. The benchmarking of the proposed review with existing work is provided in Table 5. Here, benchmarking was performed based on four components: the first includes humans’ different anatomical cancers; the second covers the different aspects of AI, such as machine learning and deep learning, cross-validation, and performance metrices; the third covers radiogenomics aspects such as conventional and deep radiomics and genotypes, and the last component covers the cohort description. The main aim of the proposed review is to cover all the aspects of radiogenomics, including the brief fundamentals of AI, and its offers with different genotypes of multiple cancers of the human anatomy. It also provides artificial intelligence’s achievements and challenges in current clinical practice. Therefore, this review reaches a different height in comparison with other studies.

## 9. Clinical Challenges and A View for the Future

The potential of radiogenomics in tumor diagnosis, prognosis, and prediction is immense; however, translations into clinical settings are slow due to several associated challenges [175]. Adopting radiogenomics practices into clinical settings needs to overcome these significant challenges. The biggest challenge is the storage, management, extraction, analysis, integration, visualization, and communication of the information generated from the myriad of available data [176]. Integrating such heterogeneous and multifactorial data in a cost-effective, standardized, and secure manner is essential. Initiatives such as the Cancer Research UK’s Stratified Medicine Program (Cancer Research UK, 2013) and the Center for Advancing Translational Science under the National Institutes of Health (NIH, 2011) have been started for better management of radiomics research in oncology [176].

The nature and variability of data are also critical standing challenges. Although vast imaging data are readily available, institutional heterogeneity (either inter- or intra-) exists because of the differences in scan protocols, hardware, and post-processing steps, thereby limiting generalizing findings [176]. Differences in image acquisition parameters, arguments, and distinctions in contrast enhancement protocols exist [127,177]. A study reported that even if the same scanning protocol was utilized for image acquisition, it still resulted in differences in radiomic feature calculations [178]. This leads to reduced reproducibility of results and impedes the development of appropriate radiogenomics models [178]. Therefore, it is necessary to implement certain standard practice guidelines to ensure the reliability and accuracy of radiogenomics studies [100].

Data availability from genetic tests is still limited and carrying out large-scale genetic testing may not be cost-effective, in addition to being challenging. The use of genetic and imaging repositories may provide a cost-effective solution [172]. Current data generated using radiogenomics are from retrospective studies with a small patient cohort; therefore, a conclusion is usually limited and cannot be generalized, warranting more extensive prospective studies [99]. Inadequacy in data stratification may result from the lack of the required volume of data leading to compromised data adaptation, optimization, and evaluation [100]. Limited-size datasets pose a high risk of overfitting models, leading to poor generalization. This can lead to incompetent decision-making with high false-positive examination rates from multicenter with different devices and imaging protocols [172,179,180]. In clinical oncology practices, it is also required to have the availability of quantitative descriptors with interpretability. This will enable a better investigation to address the heterogeneity of tumors [100,181].

A significant task involved in radiogenomics is the interpretation of the algorithms, which are highly complex, and interpretation of their inner workings is not easy; it is referred to as ‘black box’ nature [182]. This hinders the acceptance of such technology in healthcare. An easy-to-explain algorithm allows evaluation of its outputs and provides feedback for improvement. These algorithms are highly dependent on the standards available for interpreting data, which, although highly relevant, may also serve as a source for bias [183]. In various instances, outcomes of these algorithms have been proven to be more reproducible and consistent than human readings, but this induces more patient examinations and results in overdiagnosis [182]. At the same time, personalized management decisions recommended by complex algorithms may be difficult to explain, and errors and biases may become harder to detect [184].

Another significant challenge associated with radiogenomics research is the limited number of laboratories conducting such research due to the cost and challenges related to the same [172]. Further, laboratory certifications and personnel expertise are required to make the process cumbersome. Additionally, in most cases, molecular and genetic analysis usually takes place outside hospital settings, making data integration a herculean task due to the application of different genome sequencing technologies by commercial platforms [172]. This culminates in having limited imaging and genomic datasets, limiting the expansion of radiogenomics approaches.

Though AI in radiogenomics has accomplished a great deal of development in oncology, as mentioned in several studies, still there is a long way to go until oncologists regularly use it in clinics. Apart from the challenges discussed above, a significant challenge is the effective organization and preprocessing of the multi-institutional cohort of large-scale data. Though handling the multi-intuitional data is a challenging process in terms of processing, costs, and ethical clearance procedure of different institutions. However, if it is achieved intelligently then it would make the radiogenomics study the best and most clinically reliable. For example, if—due to some ethical issue—institutions cannot share their data, in this case, they can share their developed AI models and conduct tests on their cohort, and researchers can combine the models effectively and conduct further analysis. Therefore, researchers could perform their study with more robust and generalizable results. Additionally, one crucial concern of the radiogenomics study is that properly nested cross-validation must be performed to avoid overfitting, which is the common case in AI. Many studies have shown very high accuracy for cancer’s perspective, ignoring the fact that these cancers are heterogeneous, and it is not easy to learn the exact imaging phenotype by the AI model. By the way, from the future perspective, AI will become an essential tool in the radiogenomics of cancer if the challenges discussed are handled appropriately.

## 10. Conclusions

Nevertheless, in recent years, AI in radiogenomics has presented novel solutions to the current clinical challenges for treating cancers and has shown promising outcomes for personalized prognosis and treatment planning. As discussed, it is applied tremendously in various studies of cancers such as survival prediction, progression-free survival, cancer heterogeneity analysis, etc., in the era of precision medicine. However, we have noticed that certain studies have been conducted with low amounts and a lack of (i) multi-institutional data, (ii) proper cross-validation analysis, (iii) generalizable results, and (iv) robustness, thereby posing more challenges and shaking the oncologists’ confidence regarding its use in regular clinical practice. In the future, we suggest that further studies will emphasize eliminating the current limitations of AI in radiogenomics and making their AI methods more efficient for clinical purposes.

## Figures and Tables

**Figure 1 cancers-14-02860-f001:**
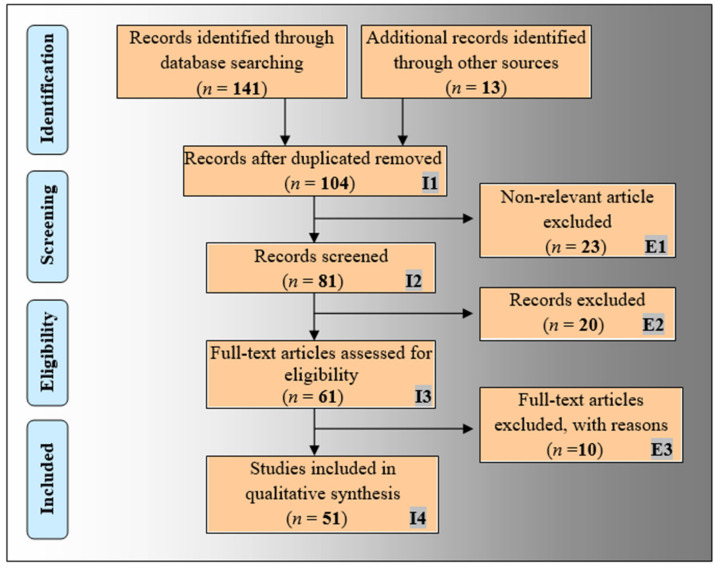
The PRISMA model.

**Figure 2 cancers-14-02860-f002:**
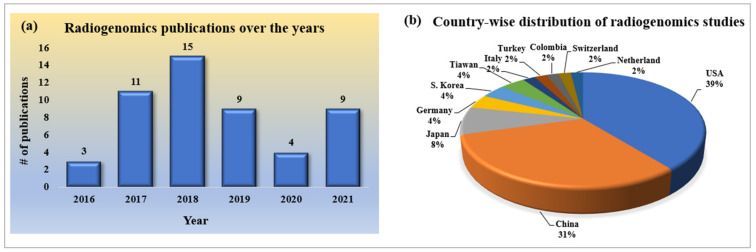
(**a**) Publication trends; (**b**) country-wise distribution of radiogenomics studies.

**Figure 3 cancers-14-02860-f003:**
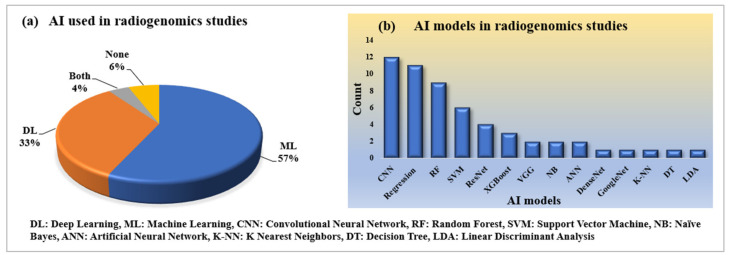
AI and its model used in radiogenomics studies (**a**) AI; (**b**) AI models.

**Figure 4 cancers-14-02860-f004:**
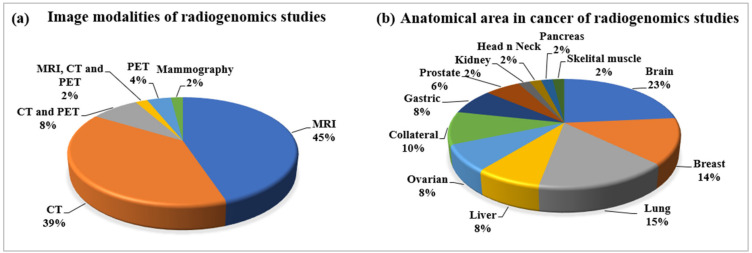
(**a**) Image modalities; (**b**) anatomical cancer in radiogenomics studies.

**Figure 5 cancers-14-02860-f005:**
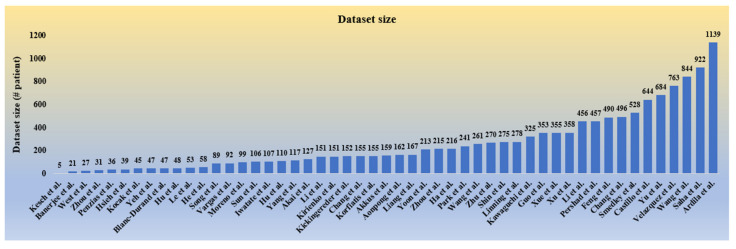
Dataset of radiogenomics studies.

**Figure 6 cancers-14-02860-f006:**
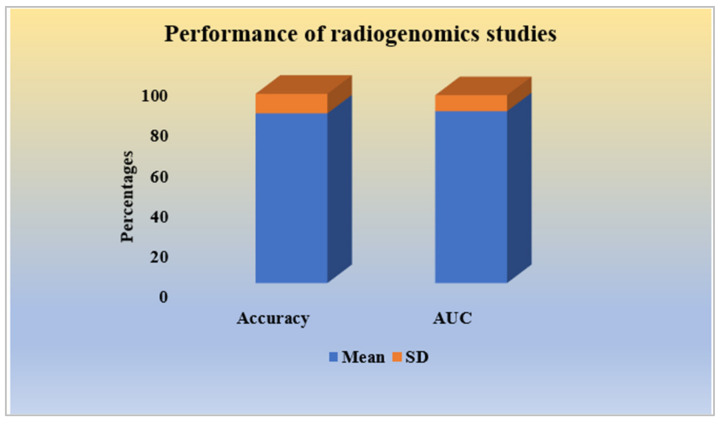
Performances of radiogenomics studies.

**Figure 7 cancers-14-02860-f007:**
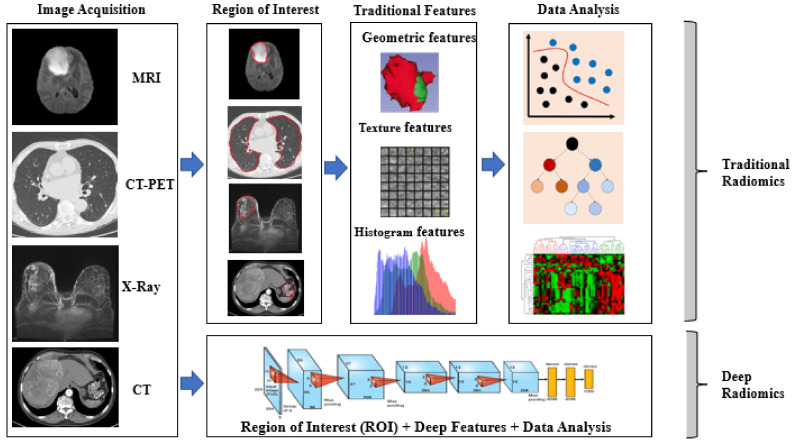
Traditional vs. deep radiomics: traditional radiomics consists of different approaches such as ROI detection, feature-extraction selection, and analysis, while deep radiomics consists of all steps in a single go [35].

**Figure 8 cancers-14-02860-f008:**
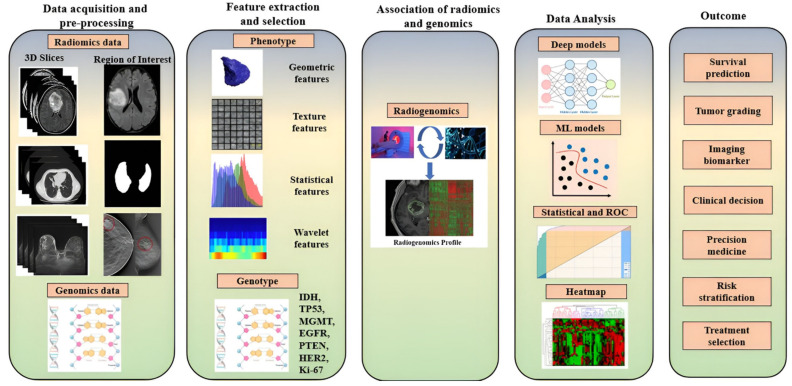
Radiogenomics pipeline of 5 stages including data acquisition (radiological imaging), preprocessing steps, features (low and high-end) extraction and selection, the association of radiomics and genomics, analysis, and finally, the radiogenomics outcome [8].

**Figure 10 cancers-14-02860-f010:**
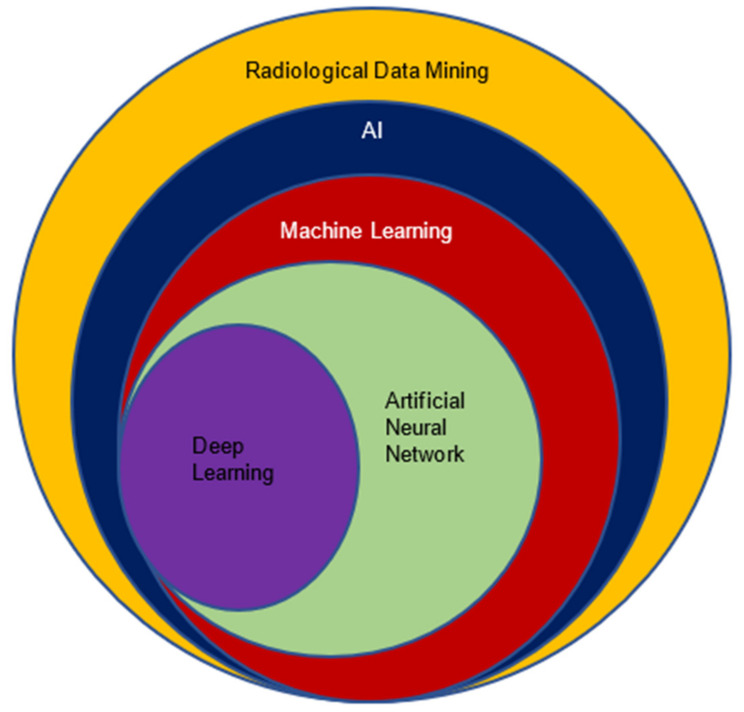
Artificial Intelligence and its subsets (machine learning; neural network; deep learning) perspective to the radiological data [119].

**Figure 11 cancers-14-02860-f011:**
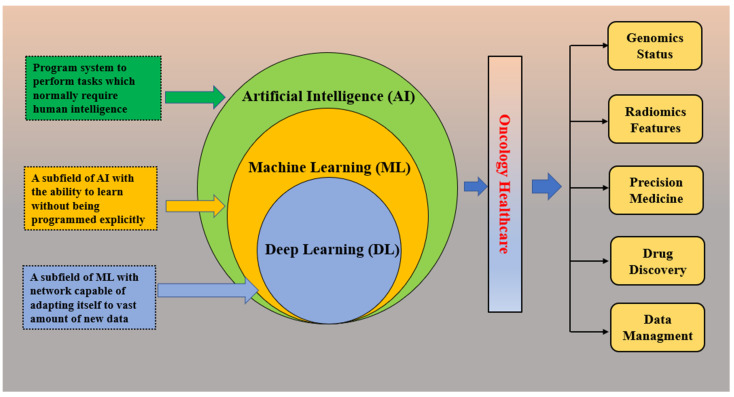
AI, its components, and the perspective of its application to oncology health care [120].

**Figure 12 cancers-14-02860-f012:**
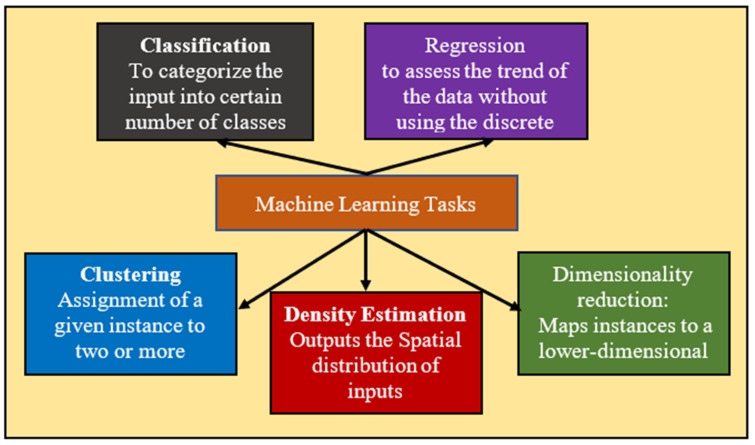
Machine learning tasks (classification, regression, clustering, density estimation, dimensionality reduction).

**Figure 13 cancers-14-02860-f013:**
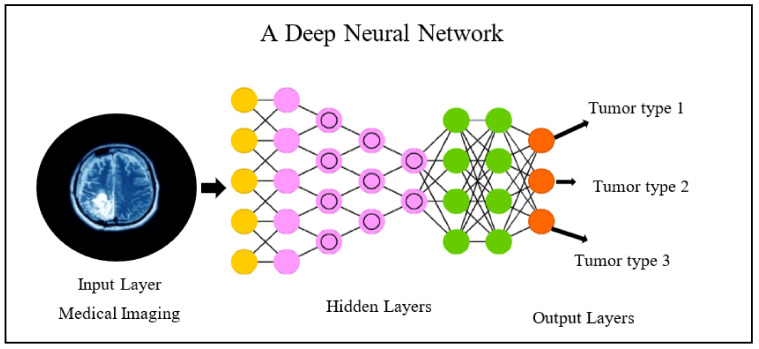
A deep neural network.

**Table 2 cancers-14-02860-t002:** AI-based model using cross-validation.

SN	Cross-Validation Type	Brief Description
1	Leave one out cross-validation	An extreme type of CV that leaves one data sample out of the total data sample, then n − 1 samples are used to train the model and one sample is used as the validation set.
2	Hold-out cross-validation	This is the usual train/test split of the dataset is a CV technique in which the dataset is arbitrarily partitioned into 2 parts of training and testing (validation).
3	k-fold cross-validation	In the k-fold cross-validation, the dataset is partitioned into k parts such that each time, one of the k parts is used as the training set and the other k − 1 subsets as the validation set.
4	Stratified k-fold cross-validation	It is a small variation of k-fold CV, in which each fold contains approximately the same strata of samples.
5	Nested cross-validation	Otherwise known as double cross-validation, in which k-fold cross-validation is employed within each fold of cross-validation often to tune the hyperparameters during model evaluation.

**Table 3 cancers-14-02860-t003:** Performance metrics of AI models.

SN	Performance Matrix	Description
1	Accuracy	It is set out as the number of correct predictions made as a ratio of all predictions made. $Accuracy=\frac{TP+TN}{TP+FP+FN+TN}$
2	Sensitivity or Recall	It is defined as the number of positive predictions made. $Sensitivity=\frac{TP}{TP+FN}$
3	Specificity	It is defined as the number of negative predictions made. $Specificity=\frac{TN}{TP+FN}$
4	Precision	It is defined as the number of correct positive results divided by the number of positive results predicted by the classifier. $Precision=\frac{TP}{TP+FP}$
6	F1-Score	It is defined as the weighted average of precision and recall. $F1-Score=\frac{2*\left(Precision*Recall\right)}{Precision+Recall}$
7	Area under ROC curve (AUC)	It is a probabilistic measure that defines how much the model is capable of distinguishing between classes.
8	Kaplan-Meier Curve	It is the visual representation of the function that shows the probability of an event at a respective time interval.
9	Mean Absolute Error (MAE)	It is defined as the average of the difference between the ground truth and the predicted values by the regression model. $\mathrm{MAE}=\frac{\sum_{i=0}^{N}\left|y_{i}-y_{i}^{p}\right|}{N}$
10	Mean Square Error (MSE)	It is defined as the average of the squared difference between the target value and the predicted value by the regression model. $\mathrm{MSE}=\frac{\sum_{i=0}^{N}{\left(y_{i}-y_{i}^{p}\right)}^{2}}{N}$
11	R^2^ (R-Squared)	It is defined as the statistical measure of fit that indicates how much total variation of a dependent variable is explained by the independent variable by the regression model. $R^{2}=1-\frac{Unexplained\;variation}{Total\;variation}$

Where *TP*—true positive; *TN*—true negative; *FP*—false positive; *FN*—false negative; $y_{i}$ and $y_{i}^{p}$ are the target variable and predicted values; *N* represents the total number of samples.

**Table 5 cancers-14-02860-t005:** Benchmarking between different radiogenomics reviews.

			Discussion of Fundamentals of Various AI Components			Radiogenomics Components			
Citation	Year	Anatomical Cancers Discussed	PM	CV	ML/DL	Conv. Radiomics	Deep Radiomics	Essential Genotypes	Dataset
Singh et al. [168]	2021	Brain	✗	✗	✓	✓	✓	✓	✓
Razek et al. [113]	2021	Brain	✗	✗	✓	✓	✓	✓	✓
Liu et al. [169]	2021	Gastrointestinal, Lung, Liver, Ovarian, Renal, Head and Neck,	✗	✗	✗	✓	✓	✓	✗
Singh et al. [170]	2021	Brain, Breast, Lung	✓	✗	✓	✓	✓	✓	✓
Wong et al. [171]	2020	Lung	✗	✗	✗	✓	✓	✓	✗
Trivizakis et al. [172]	2020	Breast, Pancreatic, Oral, Bladder, Head and neck, Rhabdomyosarcoma,	✓	✗	✗	✗	✓	✓	✗
Nougaret et al. [173]	2020	Ovarian	✗	✓	✓	✓	✗	✓	✗
Gullo et al. [174]	2020	Breast, Brain, Lung, Gynecological, Liver, Kidney, and Prostate	✗	✗	✗	✓	✗	✓	✓
Bodalal et al. [4]	2019	Brain, Lung, Breast, Ovaries, Liver, Kidney, Colorectal, Prostate	✗	✗	✗	✓	✓	✓	✗
Pinker et al. [99]	2018	Breast	✗	✗	✓	✓	✗	✓	✗
Proposed Review		Brain, Breast, Lung, Liver, Colorectal, Gastric, Prostrate, Ovarian	✓	✓	✓	✓	✓	✓	✓

PM—performance metrics; CV—cross-validation; ML—machine learning; DL—deep learning; Conv.—conventional.

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
