# Peer review of "Role of Artificial Intelligence in Radiogenomics for Cancers in the Era of Precision Medicine"

_cancers, 2022, doi:10.3390/cancers14122860_

Round 1

Reviewer 1 Report

The purpose of this review is to provide an overview of radiogenomics with the viewpoints on the role of AI in terms of its promises for computational as well as oncological aspects and offers achievements and opportunities in the era of precision medicine. The review also presents various recommendations to diminish these obstacles. This review has the strength of finding related papers on radiogenomics, deep learning models, and oncological aspects. There are some comments in this paper. 

  1. 1. Although the topic of this paper is Radiogenomics for Cancers, the explanation of radiogenomics related to cancer is relatively lacking. Much of this paper explains the concept of radiogenomics or AI. It is recommended that the talk about radiogenomics and the application of AI in cancer research be given more weight. 

  1. 2. In Section 7 (AI in radiogenomics studies of different cancer type), it is recommended to summarize representative radiogenomic studies of different cancer type in the manuscript. In this study, the text of Section 7 is too short and consists only of a list of tables. 

  1. 3. The images used in Figures 9 and 12 may be copyright issues. It is recommended that you use your own image or comment on the image copyright. 

  1. 4. Section 3.1 is repeated three times. Please correct the section number. 

  1. 5. From page 20, the sentence number is written, but the sentence number is not written up to the previous page. Also, the page number is not written until page 19. It is necessary to uniformly unify the page formatting.

Author Response

Thanks for your valuable suggestion. We have addressed your all concerns in the attached rebuttal document.

Reviewer 2 Report

This manuscript is entitled: “Role of Artificial Intelligence in Radiogenomics for Cancers in the Era of Precision Medicine.” This review manuscript provides and discusses different perspectives regarding the contemporary and inherent responsibility of AI methods in radiogenomics of cancer, including current challenges and prospects. This manuscript indicates that AI in radiogenomics has presented novel solutions to the current clinical challenges for treating cancers and has shown promising outcomes for personalized prognosis and treatment planning. There are some questions and suggestions that may need to solve as below:

1.        Some of the charts can be drawn more beautifully. For example, Figure 10 can be expressed by the scale, and the concentric circles can be modified to make the representation clearer.

2.        The data lack statistical significance, and the manuscript should be analyzed to confirm the reliability of the data.

3.        As several images were taken from the reference literature, some of the figures are too small to be distinguishable—for example, Figure 8 and Figure 12.

4.        The equations in Table 3 are taped to the dividing line, so the size of the table should be adjusted to represent the equations clearly.

Author Response

Thanks for your valuable suggestion. Hope we have addressed your concerns completely

Reviewer 3 Report

The manuscript provides an overview of AI in Radiogenomics, including the number of publications trends according to geography, number of publications, etc. The Manuscript follows a simple easy-to-follow structure. 

The authors also point out the limitations in certain studies, the challenges, and future perspectives, which in my opinion is crucial in a Review.

The manuscript is ready.

Author Response

Thanks for your valuable suggestions. Hope we have addressed your concerns completely.
